# Mental Adjustment, Functional Status, and Depression in Advanced Cancer Patients

**DOI:** 10.3390/ijerph20043015

**Published:** 2023-02-09

**Authors:** Adán Rodríguez-González, Verónica Velasco-Durántez, Patricia Cruz-Castellanos, Raquel Hernández, Ana Fernández-Montes, Paula Jiménez-Fonseca, Oscar A. Castillo-Trujillo, Miguel García-Carrasco, Berta Obispo, Jacobo Rogado, Mónica Antoñanzas-Basa, Caterina Calderon

**Affiliations:** 1Department of Medical Oncology, Hospital Universitario Central de Asturias, ISPA, 33006 Oviedo, Spain; 2Department of Medical Oncology, Hospital Universitario La Paz, 28046 Madrid, Spain; 3Department of Medical Oncology, Hospital Universitario de Canarias, 38320 Tenerife, Spain; 4Department of Medical Oncology, Complejo Hospitalario Universitario de Ourense—CHUO, 32005 Ourense, Spain; 5Department of Medical Oncology, Hospital Quironsalud, 41013 Sevilla, Spain; 6Department of Medical Oncology, Hospital Universitario Infanta Leonor, 28031 Madrid, Spain; 7Department of Medical Oncology, Hospital Universitario Clínico San Carlos, 28040 Madrid, Spain; 8Department of Clinical Psychology and Psychobiology, Faculty of Psychology, University of Barcelona, 08035 Barcelona, Spain

**Keywords:** mental adjustment, depression, advanced cancer, health-related quality of life, functional status

## Abstract

Depressive symptoms are common in individuals with advanced cancer. Objectives. This study sought to analyze the relationship between physical and functional status and depressive symptoms, and to assess the role of mental adjustment across these variables in people with advanced cancer. Methods. A prospective, cross-sectional design was adopted. Data were collected from 748 participants with advanced cancer at 15 tertiary hospitals in Spain. Participants completed self-report measures: Brief Symptom Inventory (BSI), Mini-Mental Adjustment to Cancer (Mini-MAC) scale, and the European Organization for Research and Treatment of Cancer (EORTC) questionnaire. Results. Depression was present in 44.3% of the participants and was more common among women, patients <65 years old, non-partnered, and those with recurrent cancer. Results revealed a negative correlation with functional status, and functional status was negatively associated with depressive symptoms. Mental adjustment affected functional status and depression. Patients having a positive attitude displayed fewer depressive symptoms, while the presence of negative attitudes increased depressive symptoms in this population. Conclusions. Functional status and mental adjustment are key factors in the presence of depressive symptoms among people with advanced cancer. Assessment of functional status and mental adjustment should be considered when planning treatment and rehabilitation in this population.

## 1. Introduction

Patients with advanced cancer often experience varying degrees of depression for a host of reasons, including as a reaction to the cancer diagnosis, symptoms secondary to the tumor itself, treatment, and uncertainty surrounding the risk of disease progression [1,2,3]. Estimates of the prevalence of depression in individuals with advanced cancer range from 18 to 67% [4,5,6]. According to Derogatis [7], depression is characterized by feelings of tension, worry, sadness, and irritability perceived by the patient; it is associated with decreased functional status, worse treatment compliance, longer hospitalizations, and a lower survival rate [6,8,9], prompting growing attention on the part of clinicians and researchers. 

Several psychosocial factors, including age, sex, type of tumor, and disease progression, can contribute to the development of depression in individuals affected by cancer [10,11]. In a systematic review (40 articles) involving people with cancer, females were more likely to develop depression than males; as for age, most research indicates that younger oncology patients exhibit more depressive symptoms than older ones, but thet results are inconclusive, with some studies reporting the opposite [12]. As for physical factors, people with comorbidities and other chronic conditions were at a higher risk of depression; similarly, a worse cancer stage and metastases were associated with higher rates of depression [12]. In cases of advanced cancer, younger patients and women tended to exhibit more symptoms of depression than men and older individuals [5,13].

Physical and functional status are particularly significant in advanced cancer patients, considering the extent of cancer symptoms, comorbidity with other conditions, the ease with which a person can perform daily activities, and how much help is needed for basic self-care [14,15]. Functional status has long been acknowledged as a predictor of cancer outcome [16,17] and many studies confirm its importance as a predictor of survival in advanced cases [17,18,19]. People with an impaired functional status tolerate cancer treatments worse [18,20] and experience a poorer course of disease than others with the same stage of cancer [19,20], as well as being at greater risk for suffering mood disorders, such as depression, and decreased quality of life [21,22]. Coping strategies can play a pivotal mediating role between functional status and depressive symptoms. 

In recent years, there has been a growing interest in understanding and addressing the challenges associated with coping with cancer. Mental adjustment refers to an individual’s cognitive and behavioral reactions to receiving a diagnosis of cancer [23]. Active and positive coping styles in the face of cancer correlate with better adjustment to the disease, treatment adherence, and quality of life, thereby bolstering the patient’s sense of self-efficacy and personal control [21,24,25], whereas avoidant coping with cancer, hopelessness, anxious preoccupation, or the presence of negative attitudes increase symptoms of depression in individuals with cancer and cause greater stress and difficulty in undertaking actions relating to managing their condition [26,27].

Identifying factors associated with depression can provide a reference for intervention and treatment. To the best of the authors’ knowledge, there are no studies that analyze the role of mental adjustment across physical and functional status and depressive symptoms in a large sample of Spanish patients with advanced cancer. The study objective was to examine the relationship across physical and functional status and depressive symptoms and evaluate the role of mental adjustment on these variables in advanced cancer. The hypothesis is that functional status, mental adjustment, and depression are interconnected, and that mental adjustment (as in negative and positive attitudes) play a significant role in the relationship between functional status and depression.

## 2. Materials and Methods

### 2.1. Participants and Procedure

This is a multicenter, prospective, cross-sectional study. Cases of advanced cancer were consecutively recruited from 15 medical oncology departments of different hospitals in Spain between February 2020 and June 2022. Patients were selected at their first visit to the medical oncologist who explained the diagnosis, stage, incurable disease status, and systemic antineoplastic treatment options. Eligible candidates were over 18 years of age with histologically confirmed, advanced cancer who were not eligible for surgery or other therapies with curative intent. Patients with physical conditions, comorbidity, and/or age that represented a contraindication in the opinion of the attending oncologist to receive antineoplastic treatment; who had received cancer treatment in the previous 2 years for another advanced cancer; or with any underlying personal, family and sociological, geographical, and/or medical condition that could hinder the patient’s ability to participate in the study were excluded. A total of 857 patients were enrolled; 837 were eligible, and 20 were excluded (6 did not meet the inclusion criteria; 4 met an exclusion criterion, and 10 had incomplete data), as shown in the flow diagram, Figure 1.

This research was conducted in accordance with current ethical principles and had the prior approval of the Ethics Review Committees of each institution and of the Spanish Agency of Medicines and Health Products (AEMPS; identification code: ES14042015). The study involved completing several questionnaires and collecting clinical data from the interview and medical history. Data collection procedures were similar in all hospitals and patients’ data were obtained from the institutions where they received treatment. Those who agreed to participated signed the consent form, were given instructions on how to fill in the written questionnaires, completed it at home, and handed them to the auxiliary staff at the following visit. All participants provided informed consent before inclusion. Data were collected and updated by the medical oncologist, through a web-based platform (www.neoetic.es).

### 2.2. Measures

Demographic information, including age, sex, marital status, educational level, and employment status, and questionnaires were provided by the patients in writing. The three questionnaires (Brief Symptom Inventory, Mini-Mental Adjustment to Cancer, and European Organization for Research and Treatment of Cancer Quality of Life Questionnaire) were completed at home during the interval between the first visit to the oncologist and the beginning of systemic treatment. Clinical variables pertaining to cancer such as primary tumor site, histology, recurrent cancer (yes/no), anti-neoplastic treatment, and outcomes were gathered by the medical oncologist from the medical records. 

Depression was assessed using the Brief Symptom Inventory (BSI) [7]. The questionnaire consists of six descriptions of physical and emotional complaints of depression. The depression subscale quantifies symptoms of discontentment, disaffection, and dysphoric mood, e.g., self-deprecation, anhedonia, hopelessness, and suicidal ideation. Each item is scored on a 5-point Likert scale; the score for each subscale ranges from 0 to 12, with higher scores indicating greater depression. Raw scores are converted to T-scores based on gender-specific normative data. To identify individuals with significant levels of depression, the BSI applies the clinical case-rule. According to the cut-off values recommended by Derogatis [7], patients whose T-score ≥ 63 were considered to have “probable depression”. The Spanish version of the BSI has proven good reliability and validity in the Spanish population [28]. 

Coping strategies for cancer were assessed using the Mini-Mental Adjustment to Cancer (Mini-MAC) [23]. It contains 29 items that evaluate three factors: negative attitude (anxious preoccupation and helplessness), positive attitude, and cognitive avoidance. The items are scored on a 4-point Likert scale; the higher score, the more that coping strategy is used. The Spanish version of the Mini-MAC scores had reliability estimates (Cronbach’s alpha) ranging from 0.88 to 0.9 [29].

Symptoms and functional status were probed using the European Organization for Research and Treatment of Cancer Quality of Life Questionnaire (EORTC-QLQ-C30) [30], comprising 30 items, 24 of which are aggregated into nine multi-item scales: five functioning scales (physical, role, cognitive [CF], emotional, and social); three symptom scales (fatigue, pain, and nausea and/or vomiting), and one global health-status scale. The remaining six single items assess symptoms of dyspnea, appetite loss (AP), sleep disturbance, constipation, diarrhea, and financial impact. Response choices vary from 1 (not at all) to 4 (very much). All scales’ scores are linearly transformed to a 0–100 scale. Higher scores indicate better functional status as well as more physical symptoms. Score reliability estimates for the Spanish version were 0.88–0.96 [15].

### 2.3. Data Analysis

Descriptive statistics were performed and both means (M) and standard deviations (SD) were calculated for demographic and clinical characteristics. ANOVAs were used to examine differences in depressive symptom (BSI score) as a function of demographic and clinical variables. Eta squared (η^2^) was computed to assess effect size of continuous variables. Eta-squared ranged from 0 to 1, with η^2^~0.01, η^2^~0.06, and η^2^ > 0.14 for a small, medium, and large effect size, respectively [31]. Bivariate correlations were used to evaluate the association between symptom and functional scale (EORTC), coping strategies (M-MAC), and depression (BSI). All data were inspected for normality, outliers, and the assumptions of multicollinearity and homoscedasticity [32]. Structural Equation Modeling (SEM) is capable of building, estimating, and testing theoretical models of the relationships between variables. It can substitute multiple regression and other methods to analyze the strength of correlations between individual variable indicators in a specific population [33]. In this study, the previously determined significant factors were used in the SEM to identify the relationship between symptom and functional scale, coping strategies, and depression. Standardized direct, indirect, and total effects with corresponding 95% bias-corrected confidence intervals (CI) were measure using the bootstrapping methods [33,34]. The model fit was tested by means of the normed χ^2^ value (NC; desired value < 2.0, desired significance *p*  <  0.05), goodness-of-fit index, Comparative Fit Index (CFI), Tucker–Lewis Index (TLI), Normed Fit Index (NFI) (>0.95 indicating an excellent fit), and root mean square of approximation (RMSEA; desired value < 0.06) [33]. Bilateral statistical significance was set at *p* < 0.05 for all tests. Statistical analyses were performed using the IBM-SPSS 23.0 statistical and AMOS 23.0 software package for Windows PC. 

## 3. Results

### 3.1. Demographics and Clinical Characteristics

Data from 837 participants (mean age, 65 ± 10.6) were included in the analysis after excluding missing data (n = 10). Most of the participants were men (54%); 78% were married; 52% completed junior high school; and 51% were retired or unemployed. The most common tumors were bronchopulmonary (32%), colorectal (15%), pancreatic (10%), breast (7%), and gastric (6%). Adenocarcinoma histology was the most prevalent (63%) and most cancers were diagnosed in stage IV (81%). The most frequent treatment was chemotherapy alone or combined with other treatment modalities (80%). Estimated survival was <12 months for 27% of the sample. 

Based on the cut-off values, the overall prevalence of depressive symptoms in the study population was 46%. Mean values were 63.4 ± 7.1 for depressive symptoms. One-way ANOVA denoted that there were statistically significant relationships between depressive symptoms and sex (F = 17.685, *p* = 0.001, partial eta-squared = 0.021), age (F = 2.691, *p* = 0.030, partial eta-squared = 0.013), marital status (F = 7.881, *p* = 0.005, partial eta-squared = 0.012), and recurrent cancer (F = 6.540, *p* = 0.011, partial eta-squared = 0.008); no significant differences were found for the remaining variables (see Table 1). Using a cut-off point < 75 [35] to identify people with functional problems, 41% of the participants indicated that they had difficulties in carrying out activities of daily living.

### 3.2. Correlations across Variables

Depressive symptoms correlated positively with sex (*r* = 0.144, *p* < 0.001), marital status (*r* = 0.108, *p* = 0.005), symptom scale score (*r =* 0.476, *p* < 0.001), and negative attitude (*r =* 0.534, *p* < 0.001), whereas they correlated negatively with age (*r* = −0.081, *p* = 0.020), functional scale (*r* = −0.618, *p* < 0.001), and positive emotion (*r* = −0.298, *p* < 0.001). No significant correlations were detected between cognitive avoidance and depressive symptoms (See Table 2).

### 3.3. Relationship across Symptom and Functional Scale Scores, Coping Strategies, and Depression: Path Analysis

The model exhibited excellent fit to the data (ꭕ2 = 14.718; *p* = 0.005; CFI = 0.993; NFI = 0.990; TLI = 0.982; RMSEA = 0.057 (90% CI = [0.028, 0.089]). As displayed in Figure 2, the symptom was directly and negatively associated with functional scale score (*β* = −0.76, p < 0.01); functional scale score was directly and positively associated with positive attitude (*β* = 0.22, *p* < 0.01), and negatively associated with negative attitude (*β* = −0.42, *p* < 0.01) and depression (*β* = −0.46, *p* < 0.01). Negative attitude was positively associated with depression (*β* = 0.32, *p* < 0.01) and positive attitude was negatively associated with depression (*β* = −0.15, *p* < 0.01); the more symptoms patients exhibit, the worse their functional status and the more depressive symptoms increase; furthermore, positive attitude and negative attitude mediated in the association between functional status and depressive symptoms—Figure 2. 

## 4. Discussion

This is the first study to analyze the relationship between physical and functional status, mental adjustment, and depression in a Spanish sample of individuals with advanced cancer. Physical and functional status often correlate with depression in oncological patients [8,36]. The prevalence of the symptoms of depression in this sample was 44.3%, slightly higher than Spanish patients with resected cancer (36.6%) [37], and greater than the sample of the general Spanish population (4.7%) [38]. This study suggests depressive symptoms are very high among people with cancer, particularly in those with metastasis. 

Social factors can also play an important role in patients’ emotional state [36,39]. In these series, females, younger individuals, unmarried or unpartnered people, and those with recurrent cancer displayed more symptoms of depression than males, older, married patients, and those suffering from cancer for the first time, which was consistent with the literature [5,12,13]. It would be necessary for healthcare professionals to develop and implement effective measures aimed at assessing and mitigating depressive symptoms in cases of advanced cancer, thereby improving their mental health, given the high rates of depression in advanced cancer patients.

Individuals with advanced cancer generally experience a variety of symptoms, impaired functional status, and the possibility of death [36,40,41]. In this study population, we have found that physical symptoms correlated negatively with functional status, in line with earlier findings [17,42]. Patients with advanced cancer comprise a population at particular risk, given the increase in physical symptoms and impaired functional status, which negatively impacts their psychological health and quality of life [21,22]. In the present study, 41% of the participants indicated that they had a hard time performing activities of daily living [43]. In keeping with a systematic review (n = 43 studies), between 36.7% and 54.6% of cancer sufferers report difficulties in carrying out basic, fundamental activities of daily living [43], which deteriorate further over time [44]. Functional impairment is significantly associated with longer hospital stays and worse survival [17,42]. Patients with a high risk of functional impairment may benefit from services such as home healthcare following discharge, rehabilitation, inpatient exercise programs, home hospitalization, and psychological care. These interventions can help improve the symptoms and functional status of patients [45,46,47]. 

The different coping strategies patients use to confront a scenario of functional decline yield disparate results [22,48]. How people with cancer cope is an important resource for psychological adjustment that can ease their stress and psychological distress [49,50]. Functional status was found to be associated with mental adjustment and with depressive symptoms in this study [17,44]. Negative attitudes correlated positively with depressive symptoms, while positive attitude exhibited a negative association with symptoms of depression in patients with advanced cancer. Individuals who display a positive attitude toward the disease that helps them to maintain their trust in adapting to the situation exhibited better mood, whereas those with high levels of negative attitudes, such as preoccupation, anxiety, and hopelessness, suffered more symptoms of depression [51]. The presence of negative attitudes and a declined functional state worsen the symptoms of depression. The positive correlation between negative attitude and depression is compatible with the findings of earlier studies [36,52,53], as well as with a greater sensitivity to pain [54] and shorter survival [55]. One possible explanation is that these individuals may adopt more passive behavior by not keeping follow-up appointments or ignoring symptoms of relapse [53].

### 4.1. Implications for Clinical Practice and Research

These findings have both theoretical and practical repercussions in healthcare in cases of advanced cancer. The results confirm those previously demonstrated in different cancer populations as regards the relationship between physical and functional status, mental adjustment, and depression. In practice, these data point to patients with advanced cancer as being at risk for presenting impaired functional and emotional status and that coping strategies can lessen said relationship. Mental adjustment evaluations should be contemplated when planning and treating advanced cancer. A more active, positive coping strategy can help patients better confront the challenges posed by the disease. For instance, the study by Trusson and Pilnick suggests that peer support can benefit these individuals and provide them with the chance to express their negative attitudes and concerns [56]; likewise, they can receive practical advice about self-care and improving their emotional state in these groups [56]. Therefore, the clinician should be more aware of this relationship between functional status and depression in this population, as it appears in other series of older adults with cancer in the Spanish population [57,58].

### 4.2. Limitations

This study has a series of strengths and limitations. Its greatest strength is that it examines a large population of people with advanced cancer from 15 oncology departments across a wide geographic region in Spain. Nevertheless, several limitations must be considered when interpreting the results. First, all the participants took part in the study voluntarily, which may have introduced a self-selection bias. Second, the study examined the relations between functional status, mental adjustment, and depressive symptoms, as well as analyzing differences in demographic and basic clinical variables. Other factors associated with depressive symptoms must also be probed in future research. Third, this is a transversal study; consequently, a longitudinal study would be worth undertaking to increase the power of the findings. 

## 5. Conclusions

In these series, depression was distinctly prevalent among the participants with advanced cancer. The result of this study substantiates that the patients who present a worse functional status perceive the disease as a source of uncontrolled stress with the presence of feelings of hopelessness, anxious preoccupation, anguish, and discouragement (negative attitude). They experience every change as a sign that their situation is deteriorating, resulting in them feeling worse and experiencing more depressive symptoms. In contrast, those having a more active and positive coping style, attempting to manage their situation, searching for alternatives and solutions, and maintaining their expectations as well as their confidence in being able to adapt to the challenges of the disease (positive attitude) present fewer symptoms of depression. Interventions that seek to enhance functional status and coping strategies could, ultimately, lessen symptoms of depression. 

## Figures and Tables

**Figure 1 ijerph-20-03015-f001:**
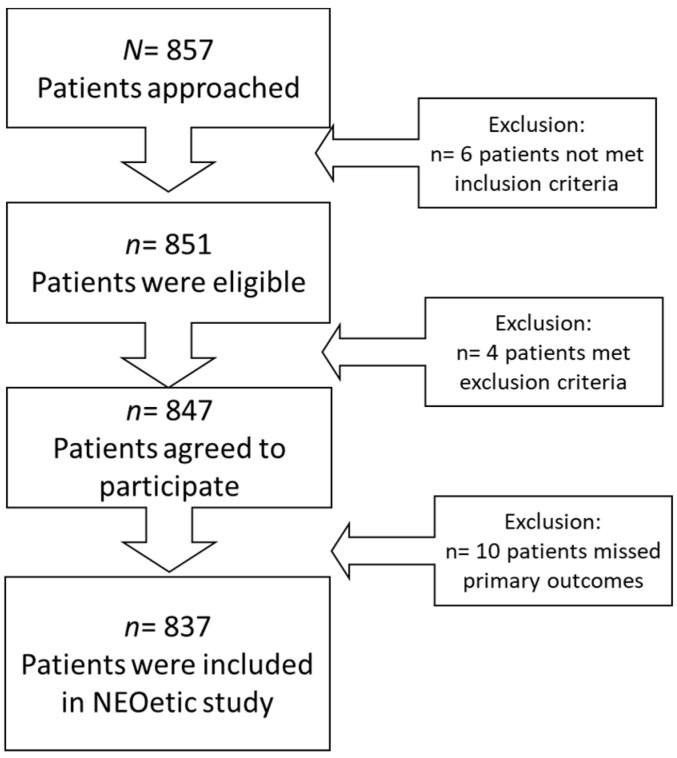
Flow diagram of the NEOetic study.

**Figure 2 ijerph-20-03015-f002:**
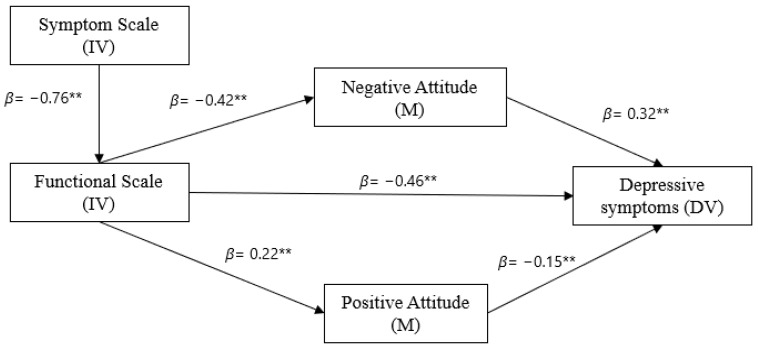
Predictive model of depressive symptoms among patients with advanced cancer. Lines present the significant pathways. ** *p* < 0.001.

**Table 1 ijerph-20-03015-t001:** Baseline characteristics (n = 837).

Variables		n (%)
Age (M; SD)	65.2 ± 10.6	
Sex	Male	454 (54)
	Female	383 (46)
Marital status	Married or partnered	653 (78)
	No partnered	174 (22)
Educational level	≤Primary school	405 (48)
	≥High school	432 (52)
Employment	No employ	422 (51)
	Employ	415 (49)
Primary tumor site	Broncho-pulmonary	266 (32)
	Colon	122 (15)
	Pancreas	83 (10)
	Breast	62 (7)
	Stomach	47 (6)
	Others	257 (31)
Histology	Adenocarcinoma	526 (63)
	Others	311 (37)
Recurrent cancer	Yes	677 (19)
	No	160 (81)
Stage	Locally advanced	161 (19)
	IV	676 (81)
Oncology treatment	Chemotherapy	670 (80)
	Others	167 (20)
Time-estimated patient survival	<12 months	224 (27)
	>12.1 months	613 (73)

Abbreviations: M, Mean; SD, Standard Deviation.

**Table 2 ijerph-20-03015-t002:** Correlations across depressive symptoms and study variables.

Variables	Depression	Age	Sex	Marital	Symptom Scale	Functional Scale	Negative Attitude	Positive Emotion	Cognitive Avoid.
Depression	1								
Age	−0.081 *	1							
Sex	0.144 **	−0.068 *	1						
Marital status	0.108 **	−0.182 **	0.126 **	1					
Symptom scale	0.476 **	−0.104 **	0.139 **	0.127 **	1				
Functional scale	−0.618 **	0.063	−0.179 **	−0.090 *	−0.764 **	1			
Negative attitude	0.534 **	0.071 *	0.051	0.037	0.298 **	−0.419 **	1		
Positive emotion	−0.298 **	−0.100 **	−0.093 **	0.014	−0.116 **	0.215 **	−0.174 **	1	
Cognitive avoid.	0.064	−0.048	−0.039	0.036	0.052	−0.043	0.277 **	0.447 **	1

* *p* < 0.05; ** *p* < 0.01. Age as a continuous variable; Sex: 0 = Male, 1 = Female; Marital Status: 0 = Married or partnered, 1 = Not partnered.

## Data Availability

The datasets generated during and analyzed during the current study are not publicly available for reasons of privacy. They are however available (fully anonymized) from the corresponding author upon reasonable request.

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
