# Peer review of "Mental Adjustment, Functional Status, and Depression in Advanced Cancer Patients"

_ijerph, 2023, doi:10.3390/ijerph20043015_

Round 1
Reviewer 1 Report
This study correlated depression symptoms, mental adjustments, and functional capacity of patients with advanced cancer. Some adjustments are required to improve the clarity and fluency of the study. The analysis was performed with a large sample, which is important. The outcomes are interesting and valid for clinical practice, but the authors must improve the report’s main message.
Introduction
Line 47 – The words “…play into depression this patient population…” is not reading well.
Line 48 – The insertion of citations 11 and 12 seems misplaced.
Line 66 – Improve the scientific writing on words “…more and more…”
Line 66 – 67 – “Mental adjustments…“ is not specific to cancer. Rewrite the sentence.
The anchoring of ideas in the last paragraph is different from the title. Additionally, it is not clear in the introduction that the study will seek to identify factors associated with depression. The first four paragraphs present the dependent variables (depression, functional status, and mental adjustments), but the linkage of all this information is not present. For example, are patients with cancer and high levels of depression expected to have reduced functional status? Could patients who present better "mental adjustments" have reduced symptoms of depression and consequent improvement in their functional status? In this sense, the hypothesis I is vague, and hypothesis II provides elements that were not properly presented in the introduction.
Methods
Line 93 – 94 “…were having physical conditions, comorbidities, and/ or age that the oncologist deemed a contraindication…” This is vague. Depression symptoms, for instance, can be affected by these variables.
Line 94 – 95 – “…other mental 94 illness or cognitive impairment…” This is also vague.
Line 87 – 98 – Some figure explaining the inclusions and exclusions (with reasons) is welcome.
Line 105 – Since authors affirm “…and did not affect patient care.”, what is the meaning of “…oncologist deemed a contraindication…” (Line 94)?
Line 111 – “…BSI, MAC, and EORTC…” ?
Line 130 – 131 – Remove the bold.
Results
Table 1 – What about the cancer location?
Table 2 – Insert the names of the variables rather than numbers in the first line.
Line 175 – The word “relationship” should be replaced here. Also, present these results in some figures with the mean and SD.
Line 205 – 214 – This information is part of the discussion, not the results.
Discussion
Line 284 – The word “symptoms” seems unnecessary here.
Conclusion
Line 291 – “In our series” ?
This section does not provide the main message of the study. Rewrite the paragraph providing a clear message to the reader.
Author Response
Reviewer #1
This study correlated depression symptoms, mental adjustments, and functional capacity of patients with advanced cancer. Some adjustments are required to improve the clarity and fluency of the study. The analysis was performed with a large sample, which is important. The outcomes are interesting and valid for clinical practice, but the authors must improve the report’s main message.
We would like to express in first place, our gratitude to this reviewer for considering that our article makes a useful scientific contribution, and especially, for their suggestions that has helped us to improve the quality of our results and their clinical implications.
Introduction
Following your suggestions, we have improved parts of the introduction, especially those you have indicated (line 47, line 66-67), the changes are highlighted in the text. We have removed reference 11 and 12. We appreciate your comments on the last paragraph, our goal is to analyze how coping strategies mediate the relationship between functional status and depression in advanced cancer patients, the first hypothesis is that all these variables are related to each other in order for mediation to occur. We have improved the paragraph that talks about the two coping strategies that can have the greatest mediational role between functional status and depression, and we have improved the manuscript's hypotheses to fit what we want to show.
Methods
Following your suggestions, we have improved the wording in methods on lines 93-95. We have included a figure with inclusion/exclusion criteria (line 90). We have also improved the wording of the study's inclusion/exclusion criteria (line 94), and we have removed the bold (line 130).
In the Results, the concept of locally advanced cancer is used to describe cancer that has grown outside the organ in which it originated, but has not yet spread to distant parts of the body. Following your suggestion, we have included the name of the variables instead of the number. We have improved the wording of the results on line 175, and we have removed the phrase from line 205 to 214, which would belong better in the discussion.
In the discussion, following your suggestion we have removed the word "symptoms."
In the conclusion, we have rewritten the paragraph to better convey our results.
The authors would like to thank the reviewer for the suggestions which have enriched our manuscript.
Reviewer 2 Report
Thank you for preparing this manuscript. My comments are as follows:
- In your introduction, there are terms you use without clear definition. The opening paragraph may benefit from additional details about depressive symptoms. For example, will all readers have a clear idea of what despondency is? Consider re-organizing the introduction, particularly the placement of the paragraph beginning on p 2 line 66.
-One of your justifications for the study is that there are no studies of Spanish patients. Perhaps it is important to know the health policy context, cultural context, or transferability of this population-- otherwise, readers may not see the importance. This also needs to be carried through in the discussion.
-Paragraph that begins p 33, line 99 could benefit from more detail, particularly because this is an international audience and hospitals may collect different information. For WC purposes, you may want to include a detailed list in supplementary materials/ appendix.
-Linking your introductory discussion about depression with variables in your measures can help with continuity.
-Description of methods and materials is largely satisfactory
-What is included in "Other" in the oncology treatment variable?
-Overall, this study is well-developed and written about clearly. There can be minor revisions to enhance the perceived contribution of the study to the larger body of literature. This rests in providing more context about the study population and why it matters, as well as a stronger description and compelling connection between depressive symptoms and oncology experiences. I believe these edits are very manageable and I look forward to seeing this manuscript in publication.
Author Response
Reviewer #2
Firstly, the researchers of this study would like to thank the reviewer for the time and effort.
In the Introduction, following the suggestions, we have described what it means to present depressive symptoms based on the description made by Derogatis about them, we have rewritten the sentence on line 45 where the relationship between the variables is described, and line 66 on the focus of the study.
Following their suggestions, we have improved the description of the materials and methods (page 3 line 33) in which the data collected by oncologists is described in more detail. All of the results collected in this study are described in the manuscript. We have also included a flow diagram.
We have improved some paragraphs on the discussion and provided information to contextualize the results of our research, which we hope will help the future reader to see the implications of the results and the transferability to other samples in the discussion section, following their suggestion (line 2037-242). There are not usually many studies with advanced cancer patients in our environment because they are fragile patients who are in a very delicate vital moment. To achieve such a large and representative sample, 15 hospitals from all over Spain have participated.
The researchers would like to thank the reviewer once again for the suggestions that have helped to improve our manuscript, we feel honored.
Round 2
Reviewer 1 Report
The authors have devoted great efforts to heeding my recommendations. Before acceptance, however, fine adjustments to the text, tables, and figures are required. As for the text, I recommend thorough proofreading of all sentences, avoiding typos. I have listed an example below, but I strongly suggest that all text be parsed.
--Line 367 – 368 – “…interventions….better [45-47]”. I really do not understand the meaning of this sentence.
--Remove “etc” and other words that decrease the quality of the study.
--Figure 1 – The quality of the figure is low. Also, the last squares show the term NEOTIC, but the legend was inserted as NEOetic.
--Table 2 - I understand how difficult it is to adjust the margins and lines of the tables. However, Table 2 is still not adequate. In the horizontal title, change 1 to Depression. On the vertical line, remove the numbers and keep all words on the same line. The Table is confusing because of the "enter" used.
Author Response
Comment: Firstly, the researchers of this study would like to thank the reviewer for the time and effort dedicated to this article. Following his suggestions, we have removed the “etc.”, and we have re-written the phrase on lines 367-68, which would now read:
“Patients with a high risk of functional impairment may benefit from services such as home healthcare following discharge, rehabilitation, inpatient exercise programs, home hospitalization and psychological care. These interventions can help improve the symptoms and functional status of patients.”
We have attached Figure 1 with higher resolution and corrected the typographical error of the term NEOetic.
Lastly, we have reviewed the tables, especially Table 2.
We greatly appreciate his observations and suggestions.